# "When will it be over?" U.S. children's questions and parents' responses about the COVID-19 pandemic

David Menendez[1]*, Rebecca E. Klapper[2], Michelle Z. Golden[2], Ava R. Mandel[2], Katrina A. Nicholas[2], Maria H. Schapfel[2], Olivia O. Silsby[2], Kailee A. Sowers[2], Dillanie Sumanthiran[2], Victoria E. Welch[2], Karl S. Rosengren[2]

1 Department of Psychology, University of Wisconsin-Madison, Madison, Wisconsin, United States of America, 2 Department of Psychology and Department of Brain and Cognitive Sciences, University of Rochester, Rochester, New York, United States of America

ꙮ These authors contributed equally to this work.
* dmenendez@wisc.edu

**Data Availability Statement:** All materials, data files and analysis scripts can be found at: https://osf.io/bmrny/?view_only=90c45df119b0496aa6dcb6f704c3cd70.

## Abstract

Parent-child conversations are important for children's cognitive development, children's ability to cope with stressful events, and can shape children's beliefs about the causes of illness. In the context of a global pandemic, families have faced a multitude of challenges, including changes to their routines, that they need to convey to their children. Thus, parent-child conversations about the coronavirus pandemic might convey information about the causes of illness, but also about how and why it is necessary for children to modify their behaviors to comply with new social norms and medical guidance. The main goal of this study was to examine the questions children ask about the COVID-19 pandemic and how parents answer them. This survey included responses from a national sample of 349 predominantly white parents of children between the ages of 3 and 12 recruited through Amazon's Mechanical Turk in United States. Parents reported that although children asked about COVID-19 and its causes (17.3%), children asked primarily about lifestyle changes that occurred as a result of the pandemic (24.0%) and safety (18.4%). Parents reported answering these questions by emphasizing that the purpose of different preventative measures was to protect the child (11.8%) or the family (42.7%) and providing reassurance (13.3%). Many parents discussed how it was their social responsibility to slow the spread of the virus (38.4%). Parents of younger children tended to shield them from information about COVID-19 (p = .038), while parents with more knowledge were more likely to provide explanations (p < .001). Our analysis shows that families not only discuss information about the virus but also information about changes to their lifestyle, preventative measures, and social norms.

**Funding:** The research reported here was supported by the Institute of Education Sciences, U.S. Department of Education, through Award #R305B150003 to the University of Wisconsin-Madison. The opinions expressed are those of the authors and do not represent views of the U.S. Department of Education.

**Competing interests:** The authors have declared that no competing interests exist.

## Introduction

With the SARS-CoV-2 (COVID-19) pandemic there has been an increase in resources targeted to parents on how to talk with their children about the situation both from official organizations [1] and other media [2–4]. Providing guidance to parents on how to talk to their children might be beneficial, as many studies show that parental talk influences children's beliefs and behaviors [5–9], including their views on the causes of illness [10] and their ability to cope with stressful situations [11–13]. However, these resources tend to focus on how to talk to children about what viruses are and how illness is transmitted while not providing evidence that children are asking questions about these topics or have any trouble understanding them. While these resources might still be useful, many children have some understanding of how illnesses work [14, 15] that they can extend to COVID-19. Instead, parents may face questions about changes in daily routines (such as working from home or attending school online) or preventative measures (like social distancing or hand washing), which might be novel for children. This information will help us determine what questions children have and potentially yield information to better guide parents on how to respond to their children.

In this paper, we examine parent-child conversations about COVID-19 from a developmental psychology perspective. Specifically, we examine parental reports of children's questions and their answers. We focus on parent-child conversations as these have been identified as critical in children's cognitive development [16, 17], the development of social cognition [18] and the development of coping skills [12]. One particular finding among this work is that children have a better understanding of phenomena when parents provide explanations rather than just stating facts [19]. Furthermore, explanations that describe the mechanism by which something works seem to be more effective at changing children's thinking [20].

Conversations between parents and children are not driven only by parents, but rather many of these conversations are started by children's questions [21]. Research in developmental psychology suggest that children's questions are key in understanding their cognition, as children use questions to fill knowledge gaps [16, 17, 22, 23]. Therefore, children's questions might give us insight into which aspects of the pandemic children want to know more about. Examining children's questions also allows us to examine whether parents are providing information to their children or evading their questions in order to shield them from information about the pandemic.

Not all parents answer their children's questions directly. In other contexts, parents have been shown to evade their children's questions in order to shield them from information they think they are too young to understand or that they think will cause negative emotions [24, 25]. Prior research shows that the likelihood that parents will engage in these conversations depends on parental attitudes [18, 26, 27]. This work suggests that parents do not always choose to engage with their children's question or do so in a way that is more comfortable to them. Therefore, we also investigate individual differences in whether parents engage with their child's question and provide explanations to them.

In this study, we focus on parental reports of children's questions and their answers. Parental reports have a long history in the study of child development. Parental reports of children's expressive vocabulary are a common and reliable measure of children's vocabulary [28]. Additionally, parental reports have been used in motor development to investigate low frequency events such as scale errors [29] and the findings of the parental report studies have been in line with those of studies that measure children's actual behavior [30, 31]. More relevant to our study, several studies have asked parents to report their children's questions either by keeping a diary [32, 33] or by asking them to report them retrospectively [34]. The results of these parental report studies have been corroborated by longitudinal studies of parent-child

conversations [33]. These parental report studies also show that children tend to ask more questions about concepts they did not yet understand, suggesting that children might have been asking these questions to fill gaps in their knowledge [34]. Overall, prior work suggests that, although imperfect, parental reports are an appropriate way to gain insights into child development as their results map on to those obtained from studies of children's actual behavior.

### Current study

The main question this study is investigating is what kind of information related to the pandemic are children seeking and what type of information are parents providing them? We examine whether children ask questions specifically about the virus, rather than about other aspects of the pandemic such as lifestyle changes. For the parental responses, we pay particular attention to whether parents attempt to shield their children from information. If parents answered their children's questions, we examine which types of explanation they provide to their children. A sub-goal was to investigate variables that might be associated with whether parents engage with their child in conversations and provide explanations. We examined parental attitudes (such as whether they think they have the knowledge to answer their children's questions) and child age as key variables that might be associated with parent's likelihood to provide explanations as prior research has shown these factors to be relevant in parent-child conversations. This study also allows us to examine how the information that children are seeking or receiving relates to demographic factors. Therefore, this study provides information about the type of information children are seeking about illness, which can enhance our understanding of the development of illness concepts. In addition, this study provides insight in how parents answer these questions and the variety of the information they provide to their children.

## Materials and method

### Participants

We recruited 500 parents over the age of 18, living in the United States, with at least one child between the ages of 3 and 12 using Amazon's Mechanical Turk. This study was approved by the Research Subjects Review Board. Participants completed the study on April 14th-15th, 2020 (roughly a month after several states and local jurisdictions declared stay-at-home orders) and were paid $2 for completing the survey. However, 151 of these individuals failed at least one of our attention checks, yielding a final sample of 349 parents (69.8% of initial sample). We had at least one respondent from 45 different states. Table 1 shows demographic information for the 349 participants.

### Measures and procedures

We set up our study such that only MTurk workers that were in the United States, had a parental status indicating they were parents and had a lifetime approval rating greater than 90% could access our study. The first two conditions helped us ensure that potential participants that saw our study met our eligibility criteria. The last condition is typical in research studies using MTurk as it improves the quality of the data (as only workers that have done good work previously are able to see the study). Before the survey, participants completed a screener verifying that they were a parent, had a child between the ages of 3 and 12, and resided in the United States. These questions were mixed among other distractor questions so that participants did not know which answers they needed to provide in order to access the survey.

**Table 1. Demographic information of participants.**

| Characteristic | No. (%) |
|---|---|
| Gender | |
| Men | 140 (40.1) |
| Women | 208 (59.6) |
| Did not respond | 1 (0.3) |
| Race/ethnicity | |
| White | 280 (80.2) |
| Black or African American | 27 (7.7) |
| Asian or Asian American | 17 (4.9) |
| Hispanic or Latinx | 10 (2.9) |
| Bi- or multi-race | 7 (2.0) |
| Other | 2 (0.6) |
| Did not respond | 6 (1.7) |
| Parent education level | |
| Some high school | 3 (0.9) |
| High school degree | 30 (8.6) |
| Some college | 58 (16.6) |
| Associates degree | 52 (14.9) |
| Bachelor's degree | 147 (42.1) |
| Master's degree | 48 (13.7) |
| Doctoral level degree | 11 (3.1) |
| Age, mean (SD), years | 38.9 (7.5) |
| Mean (SD) no. of adults in the household | 2.1 (0.7) |
| Mean (SD) no. of children in the household | 1.9 (1.00) |
| Focus child age, mean (SD), years | 7.8 (3.1) |
| Child gender | |
| Boy | 185 (53.0) |
| Girl | 163 (46.7) |
| Did not respond | 1 (0.3) |
| MacArthur Perceived Social Status, mean (SD) | 5.3 (1.7) |
| Parent biology knowledge, mean (SD) | 4.5 (1.2) |
| Child biology knowledge, mean (SD) | 3.4 (1.4) |
| Anxiety, mean (SD) | |
| Parent | 5.8 (2.9) |
| Child | 3.4 (2.8) |
| Family | 5.3 (2.5) |
| Stress, mean (SD) | |
| Parent | 6.3 (2.6) |
| Child | 3.8 (2.7) |
| Family | 5.8 (2.3) |
| Coping, mean (SD) | |
| Parent | 7.0 (2.0) |
| Child | 7.4 (2.0) |
| Family | 7.0 (1.8) |
| Mean (SD) percent of conversations started by the child | 39.7 (30.1) |

Participants who passed the screener were presented with the consent form and allowed to continue to the survey. Our survey contained 59 questions and was administered through Qualtrics® (Provo, UT). Below we provide information on the sections we discuss in this paper, and the full survey can be found here: https://osf.io/bmrny/?view_only=90c45df119b0496aa6dcb6f704c3cd70.

First, we asked parents to report the age and gender of up to five of their children. Then, parents indicated which child between 3 and 12 they were focusing on during the survey (they were instructed to focus on the child with the most recent birthday).

**Coping and anxiety scale.** We asked parents to report "how well [they] (personally) are coping with the situation surrounding COVID-19?" using a 0 (not at all well) to 10 (very well) scale. They were asked the same question about their family and their child. We also asked them "how worried [they] (personally)/ [their child]/ [their entire family] are regarding the situation surrounding COVID-19?" using a 0 (not at all) to 10 (very) scale. We also asked them to report how anxious they, their child, and their family feel using the same scale. An exploratory factor analysis suggests that the worry and anxiety measures load onto the same factor. Information on this factor analysis is presented in S1 File.

**Children's questions.** We have two measures of children's questions: (1) Parents reported up to *three questions* their children asked them about the situation surrounding COVID-19 (we refer to these as **reported questions**), and (2) Parents answered whether their children had ever asked: "Why (we need to wash our hands/ we are not allowed to go outside/ we should use hand sanitizer/ they cannot go to school/ they cannot go to the park/ we need to stay away from other people (i.e., social distancing))?" (we refer to these as **targeted questions**). Parents then described how they responded to their children's questions. The reported questions allowed us to examine the variety of topics children ask about, as parents could report any of the questions their child asked. However, because of the variety of topics and question structure, it is difficult to examine how parents respond to these questions beyond examining whether they respond or evade the questions. The targeted questions provide a smaller set of questions with similar structures and topics. This allowed us to examine the type of information that parents are providing to children.

We then asked parents if they had "come across any advice in media reports or other sources that discusses how a parent (or other adult) should answer children's questions regarding the COVID-19 situation?" If they answered "yes," we asked them to tell us about the report and if the information they received from the report was reflected in the answers they provided to their children's questions (and if so to explain). We then asked parents what percentage of the COVID-19 related conversations were started by the child.

Finally, we asked parents if they had noticed "changes in the types of questions [their] child asks about health or illness in the last month?" If they answered "yes," we asked them to explain.

**First attention check.** To ensure that parents were fully reading the questions, we presented them with a question about which news channel they usually watch, but at the end of the question we asked them to select "slightly disagree" regardless of which channel they actually watch. The response options were all real news channels with "slightly disagree" as the only non-news-related option. Participants who did not select slightly disagree were excluded from the analyses. This type of attention check is commonly used with online samples to screen for random responders.

**Shielding.** We asked parents to report whether they shielded their "child from information about the COVID-19 situation." The options we provided were (in the order presented): "I do not shield my child from information about the COVID-19 situation," "I think that the COVID-19 is no different than the seasonal flu," "I think we are making too big a deal about

the virus," "I do not want to worry them," "I do not think they are able of coping emotionally with it," and "I do not think they are old enough to understand what is going on." Parents could choose multiple reasons.

**Knowledge.**   Parents rated their own and their child's understanding of biology using a 1 (Far above average) to 7 (Far below average) scale. Parents answered if they "feel like [they] have enough knowledge or information to answer [their] child's question about the COVID-19?" using a 1 (Strongly disagree) to 7 (Strongly agree) scale.

**Coping.**   We had two open-ended questions where parents could write about the coping strategies they or their family are using, or any other information about how their family is handling the situation. For the results of these coping strategies see S1 File. For the coding scheme for these responses see Tables B and C in S1 File.

**Second attention check.**   We asked parents to report the age of the child they were focusing on during the survey. This question meant to uncover if parents were lying about the age of the child when they first reported their children's ages at the beginning of the survey (as it might be difficult to remember the exact age provided). This question also served to make sure that parents with multiple children were consistent on which child they were answering the survey about. We allowed a one-year difference between the age reported at the beginning and end of the survey to account for errors in typing. Participants who had a discrepancy larger than one year were excluded from the analyses.

**Demographics.**   Finally, parents reported demographic information including age, gender, race/ethnicity, occupation, education level, subjective SES [35], state of residence, and number of adults and children in the household.

## Qualitative coding

We coded the topic and content of participants' open-ended responses. We created the coding scheme using the open-ended responses from participants who did not pass the attention checks. Two researchers independently read the responses and came up with themes. Then, they met to discuss the themes. This coding scheme was then used for the responses of the participants that passed the attention checks. As they applied the coding scheme to the actual data, coders were asked to note if there were any changes that needed to be made to the original coding scheme. The coders did not find items that did not fit with the original categories derived from the excluded data, suggesting that our coding scheme captured the data reasonably well (the only apparent difference was higher proportion of responses coded as "other" in the discarded data). It is worth mentioning that the majority of our coding categories were mutually exclusive so that questions and answers could only be coded into one category. There were some categories that allowed multiple codes. For examples, regardless of the category a question was coded into, every question could also be coded as requesting an explanation. Similarly, regardless of the content of the answer, every answer could also be coded as providing reassurance to the child.

Two independent coders coded all of the responses. The reliability, measured through percent agreement and Cohen's kappa, was deemed appropriate for all categories (kappa values above.6 represent substantial agreement [36]). See Tables 2–5 for the coding schemes and reliability measures for each category. All disagreements were resolved through discussion. Throughout the paper, we report quotes from the written responses of our participants. These quotes have not been edited and no grammar or spelling errors have been corrected. This is to leave the data we collected intact.

**Statistical analysis.**   In order to examine whether a topic varied as a function of family factors, we fitted a generalized linear mixed-effects model (with a binomial link function)

**Table 2. Coding of children's questions.**

| Code | Description | Example | Kappa | No. (%) |
|---|---|---|---|---|
| Related to the virus | Questions about the nature of the coronavirus or the disease in general. | "What is COVID-19?" | 0.78 | 168 (17.3) |
| | | "How does coronavirus spread?" | | |
| | | "When will it be gone?" | | |
| Safety of child, family, or friends | Questions about how the coronavirus may affect the child, family, or friend's safety. | "Are my friends going to die?" | 0.90 | 178 (18.4) |
| | | "Am I at risk of getting sick?" | | |
| | | "Is the world ending?" | | |
| Preventative measures | Questions about any preventative measures that have been recommended to slow the spread of COVID-19. | "Why do I have to wash my hands?" | 0.92 | 32 (3.3) |
| | | "Why do I have to use hand sanitizer?" | | |
| | | "Why are people wearing masks?" | | |
| Lifestyle changes | Questions about lifestyle changes that came as a result of federal and state orders to slow the spread of COVID-19. | "Why can't we leave the house?" | 0.80 | 233 (24.0) |
| | | "Is the Easter bunny coming?" | | |
| | | "When can we go places?" | | |
| Related to school or work | Questions related to school or work. | "When can I go back to school?" | 0.96 | 185 (19.1) |
| | | "How long will you be working from home?" | | |
| Interpersonal interaction | Questions about interacting with other people (i.e. friends, family, teachers, etc.) | "When can I see my friends?" | 0.92 | 134 (13.8) |
| | | "Why can't I go to birthday parties?" | | |
| Other | Any question did not fit the categories listed above. | "No panic" | 0.64 | 42 (4.3) |
| | | "Eat healthy food" | | |
| Request explanation* | Any question that request an explanation from the parent. Typically, "why" or "how" questions | "Why can't I go outside?" | 0.96 | 342 (35.3) |
| | | "How does COVID -19 spread?" | | |

* indicates categories that could be coded with other categories.

predicting whether the code was present. For all the models in the paper, unless otherwise specified, we included the child's biological knowledge, the child's age, child's coping, and child's stress (the average of the ratings of child anxiety and worry) and their subjective SES as predictor variables. For models of parents' answers, we also included parent's biological knowledge and the parent's ratings of having enough knowledge to answer the child's questions. Because parents reported multiple questions, we also included by-subject random intercepts. We used a Kenward-Rogers approximation to calculate the degrees of freedom and we set our alpha level to .05 (two-sided) for all tests. Given that we use logistic models throughout the paper, we report the results of the Hosmer and Lemeshow goodness of fit test [37]. This test compares the value the model predicts for each participant with that participant's actual score to determine how well the model fits the data. This test provides a $\chi^2$ statistic and a p-value, where rejecting the null hypothesis suggests that the model does not fit the data well. Results of all of the Hosmer and Lemeshow tests can be seen in Table 6. Given that the test uses p-values, it is more likely that the model does not fit the data as the number of participants increases. Thus, we do not focus on this goodness of fit statistic, but rather examine whether including each variable adds to the explanatory power of our model by conducting a Wald's test for each model. This test compares a model with all variables to a model with the same variable except the one being tested to examine whether adding the variable increases the explanatory power of the model. All analyses were conducted in R [38] using the *lme4* package [39]. We used the *generalhoslem* package for the Hosmer and Lemeshow test [40] and the *car*

**Table 3. Coding of parents' responses about changes that they noticed in their children's questions.**

| Code | Description | Example | Kappa | No. (%) |
|---|---|---|---|---|
| No | Parent reported not noticing any changes. | "No" | 1.00 | 269 (77.1) |
| Questions about health | Parents reporting their child asking more biology-focused questions, such as the origin of illness, death, the body, etc. | "He is more curious about how the body works and how people can get sick" | 0.88 | 37 (10.6) |
| | | "They ask more about 'the virus'" | | |
| Questions about safety | Parents reported their child asking more questions about their well-being or the well-being of others | "They are concerned family members will become sick" | 0.73 | 11 (3.1) |
| | | "she started asking about the cat and expressing concern when she heard about a tiger getting sick" | | |
| Questions about lifestyle changes and preventative measures | Parents reported their child asking questions about enforced lifestyle changes and preventative measures. | "She is inquiring more about the virus and why we need to wash our hands and stay away from people, especially loved ones right now" | 0.61 | 18 (5.2) |
| | | "mostly her questions have focuses around when the lockdown will end and when she can see her friends and go back to school" | | |
| Number of questions | | | 0.93 | |
| Increase | Parents report an increase in the number of questions without mentioning changes in the content | "They ask me more questions about the situation than before" | | 3 (0.9) |
| | | "She asks much more questions than usual and very specific" | | |
| Decrease | Parents report a decrease in the number of questions without mentioning changes in the content | "She learned what she wanted in the beginning. Questions about the virus is now close to 0" | | 4 (1.1) |
| Other | Responses that did not fit into any of the above categories. | "no panic" | 0.72 | 17 (4.9) |
| | | "We talk about questions as they come up and need answers" | | |

package for the Wald test [41]. All materials, data files and analysis scripts can be found at: https://osf.io/bmrny/?view_only=90c45df119b0496aa6dcb6f704c3cd70.

## Results

### Reported questions

Overall, parents reported 969 questions that their children had asked them (on average each parent reported 2.77 questions). Parental reports suggest that children's questions tended to be about lifestyle changes that occurred as a result of the pandemic, such as not being able to visit friends or go to the park. The reported questions also asked about the child's school or their parent's work, their personal safety or the safety of their friends or family members, COVID-19 or viruses more generally, interpersonal interactions, and the preventative measures recommended to combat COVID-19. See Table 2.

We found that children whose parents reported they were coping well with the pandemic were more likely to report questions about the virus than parents who reported their children were coping less well, $OR = 1.18$, $\chi^2(1, N = 338) = 5.51$, $p = .019$. We also found that children whose parents reported they were stressed about the pandemic were more likely to report questions about the virus, $OR = 1.07$, $\chi^2(1, N = 337) = 6.14$, $p = .013$, more likely to report questions about the safety of themselves or loved ones, $OR = 1.06$, $\chi^2(1, N = 337) = 6.67$, $p = .010$, less likely to report questions about lifestyle changes, $OR = 0.94$, $\chi^2(1, N = 338) = 10.57$, $p = .001$, and less likely to report questions about interpersonal aspects, $OR = 0.94$, $\chi^2(1, N = 338) = 6.42$, $p = .011$, than parents who reported their children were less stressed. We found that parents of older children were more likely to report questions about school or work,

**Table 4. Coding of parent's responses to reported questions.**

| Code | Description | Example | Kappa | No. (%) |
|---|---|---|---|---|
| Authority | | | 0.66 | |
| Parental | When parental authority is used to answer the question. | "that's a decision their parents made up" | | 7 (0.7) |
| | | "If you follow what I say you won't get sick" | | |
| Government or official | When the authority of government officials or bodies, medical professional, or local authorities is used to answer the question. | "…wait until the doctors tell us it's okay to" | | 35 (3.6) |
| | | "As soon as the government says it's safe." | | |
| | | "We have to stay at home until the authorities say otherwise." | | |
| Religious | When the authority of religious figure or higher power is used to answer the question. | "…and prayed up God will protect us" | | 2 (0.2) |
| | | "God only knows about it" | | |
| Other | When there is an authority that has not been explicitly specified or multiple authorities are mentions. | "In a few months when they lift the restrictions" | | 14 (1.4) |
| | | "When they announce on the news that everyone can go out" | | |
| No Explanation | Responses that do not provide an explanation. | "I'm not sure it might be fall" | 0.43 | 509 (52.7) |
| | | "Yes" | | |
| | | "We don't know yet." | | |
| Explanation | | | 0.63 | |
| Realistic | Responses that provide an explanation that can be described as scientifically, medically, or historically reasonable. Common sense rational explanations are also included. | "Most likely, it would just be like a minor flu for us since we're so healthy…" | | 347 (36.0) |
| | | "It is a bad germ like the flu. You need to wash your hands and cover your sneezes" | | |
| Supernatural | Responses that provide a supernatural explanation or personified the virus. Any reference to religious activities, practices, beliefs or personas that are typically seen as religious are included. | "Yes, the Easter Bunny just hopped over the coronavirus…" | | 13 (1.3) |
| | | "A tiny invader that attack's a person's body…" | | |
| | | "The virus is very busy traveling around" | | |
| Other | Responses that did not fit into any of the above categories. | "Obama knew and tried to warn us about it" | 0.80 | 38 (3.9) |
| | | "avoid non-cooking food" | | |
| | | "When can I see my friends again?" | | |
| Reassurance* | If the parent tried to reassured the child. | "…we will be okay…" | 0.64 | 128 (13.3) |
| | | "…don't worry…" | | |
| | | "…it will get better" | | |
| | | "…we have nothing to worry about" | | |
| Religious information* | Responses that reference any type of religious concepts (i.e. actions, figures, anecdotes, etc.). | "…still worship and learn about God" | 0.83 | 7 (0.7) |
| | | "God will protect us" | | |
| | | "I pray soon baby" | | |
| | | "just have to hope and pray" | | |
| Match* | | | 0.68 | |
| Evades | The parent does not answer the question. | Q: "are you going to die" | | 20 (2.1) |
| | | A: "oh, don't talk like that" | | |
| Answers and redirects | The parent provides a short response and then elaborates on something different (not related to the question). | Q: "will daddy die" | | 25 (2.6) |
| | | A: "I'm not sure, it's important we stay home and wash our hands" | | |
| Answers Question | The parent answers the child's questions directly. | Q: "where can you get it?" | | 883 (91.5) |
| | | A: "anywhere" | | |
| | | Q: "When will it be over?" | | |
| | | A: "I'm not sure, soon" | | |

Some of the categories have low kappas primarily due to low frequency, but overall kappa for the categories is 0.62.

* indicates categories that could be coded with other categories.

**Table 5. Coding for parent's responses to targeted questions.**

| Code | Description | Example | Kappa | No. (%) |
|---|---|---|---|---|
| Authority | | | 0.71 | |
| Parental | When parental authority is used to answer the question. | "It's what we have to do to" | | 19 (1.6) |
| | | "We just can't right now." | | |
| Government or official | When the authority of government officials or bodies, medical professional, or local authorities is used to answer the question. | "the city close it because of coronavirus" | | 59 (4.9) |
| | | "Because our State governor gave the entire State a stay at home order because it's more safe." | | |
| Other | May be used when there is an authority that has not be explicitly specified or there is a combination of multiple authorities. | "Because they want to keep everyone healthy." | | 119 (9.9) |
| | | "…but we will be ablest go as soon as they tell us we can go." | | |
| Supernatural Explanation | Responses that provide a supernatural explanation or personified the virus. Any reference to religious activities, practices, beliefs or personas that are typically seen as religious are included. | "The tiny invader lives outside and nobody can see it…" | 0.81 | 14 (1.2) |
| | | "Because evil forces rule our world." | | |
| | | "to get rid of invisible germs that could cause covid-19" | | |
| Self-Protection | | | 0.75 | |
| Child Protection | The response includes an answer that relates to keeping the child safe. | "…we don't want you to get sick too" | | 142 (11.8) |
| | | "because you have to be safe" | | |
| Family Protection | The response includes an answer that relates to keeping other family members safe. | "we don't want grandma or grandpa getting sick" | | 4 (0.3) |
| | | "because in that form the virus cant get around our family" | | |
| Both (Child + Family) | The response includes an answer that relates to keeping both the child and the family safe. | "It helps keep us safe from getting it…" | | 511 (42.4) |
| | | "Because we don't want to catch the bad cold" | | |
| Social Responsibility | The response promotes awareness of other people's health or well-being. | "We don't want to spread the virus" | 0.82 | 463 (38.4) |
| | | "we want to protect others" | | |
| Contradict Social Norms | | | 0.88 | |
| Contradict and Set Boundaries | The parent negates the questions but sets boundaries in terms of actions. | "We are allowed outside in our yard, but we cannot go other places…" | | 47 (3.9) |
| | | "We can go outside, we just have to stay away from other people" | | |
| Contradict | The response contradicts the question without setting any boundaries. | "You can go. Please go and get out of the house." | | 6 (0.5) |
| | | "Well we can go outside and go to the park and fishing and stuff…" | | |
| Other | Responses that did not fit into any of the above categories. | "We always wash our hands" | 0.74 | 48 (4.0) |
| | | "6 ft." | | |

$OR = 1.08$, $\chi^2(1, N = 338) = 6.78$, $p = .009$, and less likely to report about lifestyle changes, $OR = 0.92$, $\chi^2(1, N = 338) = 9.65$, $p = .002$, than parents of younger children.

Additionally, as can be seen in Table 2, about a third of the reported questions ($n = 342$, 35.3%) requested an explanation from the parents, meaning that they were "why" or "how" questions. However, the proportion of questions that requested an explanation varied by the content of the question, $\chi^2(5, N = 954) = 107.52$, $p < .001$. A greater proportion (84.4%) of questions about preventative measures requested explanations. While a smaller proportion of questions about safety (10.8%) and school or work (25.0%) requested explanations. Parents of

**Table 6. Hosmer and Lemeshow goodness of fit test results AIC, and BIC for all the models.**

| Outcome variable | $\chi^2(8)$ | p-value | AIC | BIC |
|---|---|---|---|---|
| Reported children's questions | | | | |
| Virus | 121.17 | < .001 | 864.0 | 898.1 |
| Personal safety | 106.45 | < .001 | 904.0 | 938.1 |
| Preventative measures | 2.15 | .976 | 257.5 | 291.6 |
| Lifestyle changes | 8.82 | .358 | 1054.9 | 1089.0 |
| School/Work | 5.84 | .665 | 940.5 | 974.6 |
| Interpersonal | 6.97 | .539 | 777.0 | 811.1 |
| Request explanation | 77.99 | < .001 | 1111.7 | 1145.7 |
| Parent's answers reported questions | | | | |
| Explanation | 86.88 | < .001 | 1058.8 | 1112.1 |
| Parent's answers targeted questions | | | | |
| Protection | 132.04 | < .001 | 1548.6 | 1604.6 |
| Social responsibility | 88.78 | < .001 | 1408.4 | 1464.4 |
| Shielding | 5.27 | .729 | 464.9 | 499.5 |

older children were less likely to report questions that requested explanations than parents of younger children, $OR = 0.73$, $\chi^2(1, N = 334) = 45.49$, $p < .001$.

Only a small portion of parents ($n = 80$, 22.9%) said that their child's questions had changed in the last month. When asked to describe these changes, parents reported that their child was asking more questions about health concepts, lifestyle changes and preventative measures, and the safety of themselves and loved ones. Some parents also reported just an overall increase in the number of questions ($n = 3$, 0.9%) while others reported a decrease in the number of questions ($n = 4$, 1.1%). See Table 3.

Next, we examined how parents answered their child's questions. Parents reported 965 responses, but we eliminated responses where parents either did not report a question or their answer to a question. Overall, we obtained 928 complete sets of questions and responses. Parents tended to directly answer their children's questions ($n = 883$, 91.5%), which might be related to the fact that parents tended to rate themselves as having enough knowledge to answer their children's questions ($M = 5.61$, $SD = 1.05$). However, in some cases ($n = 25$, 2.6%), parents partially answered the question and then redirected the child to a different topic. For example, a 12-year-old girl asked, "whose going to take care of me if something happens to you?" and her mother responded, "we are! because nothing will happen to us! we are taking care of ourselves to ensure we stay healthy." Finally, in a small number of instances ($n = 20$, 2.1%) parents completely evaded their child's questions. For example, when a 3-year-old boy asked, "Why can't i go out to play?" the father evaded the question and simply answered "We are going to play at home!" In this case, the father evaded the child's request for an explanation about why they cannot play outside by telling the child they were going to play inside. See Table 4 under match category.

Parents tended to answer their child's questions without providing an explanation ($n = 509$, 52.7%). See Table 4. One hundred and thirteen (11.7%) of these answers without explanations were simple "yes," "no," "maybes" or "I don't know," while remaining answers provided facts without explanations ($n = 396$, 41.0%). When parents provided an explanation in their answers ($n = 360$, 37.3%), the majority of these explanations were realistic, but some parents mentioned supernatural elements or personified viruses ($n = 13$, 1.3%). An example of a realistic explanation is provided by the mother of a 5-year-old boy who answered the question "Why can't I go to the park?" by saying "Because there could be germs there." A supernatural explanation was

provided by the mother of a 4-year-old boy who answered, "Can we go to the toy store" by saying "We can go to the toy store when the bad cold goes to time out." In this example, the mother is personifying viruses as agents that can be put in time out. In 58 responses (6.0%) parents answered the questions by evoking some type of authority. Some parents evoked the government or other official authority, their own authority, religious authority, or some other form of authority (see Table 4 for frequencies and percentages). For example, the father of a 7-year-old boy evoked government authority when they answered, "when can I go back to school?" by saying "As soon as the government says it's safe." Many parents used this opportunity to comfort or reassure their child. Some parents also mentioned religious information in their responses. One example of parents mentioning religious information was when a 12-year-old boy asked "Will I catch it?" and the mother responded "As long as we are safe and prayed up God will protect us."

We examined if any of our variables were associated with whether parents provided explanations to their children. In addition to the variables included in all models, we added whether the question requested an explanation. We also included by-subject random slopes for the requesting explanations. We found that parents were 12.09 times more likely to provide an explanation when the child's question requested one, $\chi^2(1, N = 334) = 83.91$, $p < .001$. This finding suggests that, although some parents who did not provide an explanation even when children asked for one, parents were generally providing the information that children requested. Additionally, as parent's rating of having enough knowledge increased, the likelihood of providing an explanation increased, $OR = 1.59$, $\chi^2(1, N = 334) = 12.60$, $p < .001$.

## Targeted questions

Parents provided 1205 responses to the targeted questions. Unlike the reported questions we discussed above, the form and content of these targeted questions was the same for all participants. Critically, all of the targeted questions were **why** questions which requested an explanation from the parents. Given that the form and content is the same for all parents and consistent across questions, we could develop a coding scheme that examined more in-depth the types of responses and explanations parents provided to their children. Out of our 349 parents, two-hundred and sixty-five (75.9%) reported that their child had asked why they needed to social distance, 253 (72.5%) reported that their child had asked why they could not go to school, 239 (68.5%) reported that their child had asked why they could not go to the park, 213 (61.0%) reported that their child had asked why they could not go outside, 134 (38.4%) reported that their child had asked why they needed to wash their hands, and 101 (28.9%) reported that their child had asked why they needed to use hand sanitizer.

The majority of the answers ($n = 657$, 54.5%) explained that these measures were needed to keep *the child* safe ($n = 142$, 11.8% of total answers), some said they were to keep *the rest of the family* safe ($n = 4$, 0.3% of total answers), and some said they were to keep *the whole family* safe ($n = 511$, 42.4% of total answers). See Table 5. For example, the father of a 9-year-old girl answered a question about why their child cannot go to school with "We don't want you to get sick." Many parents explained that those measures were necessary because there is a social responsibility to keep others safe. For example, the father of a 7-year-old boy answered the question "Why we need to stay away from other people (i.e. social distancing)?" by saying "Staying away from most people will help keep everyone from getting sick." Some responses ($n = 197$, 16.3%) evoked authority when answering these questions. Of these, some responses appealed to their parental authority, the government or some other official authority, and many used other forms of authority (see Table 5 for frequencies and proportions). A small number of responses ($n = 53$, 4.4%) contradicted these protective measures. In the majority of

these 53 responses the parents set some form of boundary ($n$ = 47, 3.9% of total answers), such as saying that they can go to the park but they can't touch anything there. Some of the responses ($n$ = 6, 0.5% of total answers) explicitly said that these measures were not necessary, such as when the mother of a 9-year-old boy answered the question "Why we need to stay away from other people (i.e. social distancing)?" with "You do not need to stay away from people;". Some responses provided a supernatural explanation or personified the coronavirus ($n$ = 14, 1.2%).

One interesting finding is that the majority of the parental explanations were about protecting the family or upholding their social responsibility. It could be that parents used these explanations differently depending on the behavior being asked about. For example, parents could explain that washing hands is for the protection of the child, but that they can't go outside because it is their responsibility to slow the spread of the virus. We explored this possibility by examining whether the distribution of responses changed depending on whether parents were answering questions about hygiene (washing hands or using sanitizer) or staying home (social distancing, not going outside, not going to the park, or not going to school). We added question type (hygiene or staying inside) to our predictor variables. We also included by-subject random intercepts and by-subject random slopes for question type. The random effects were initially allowed to correlate but due to convergence issues the model we report for social responsibility did not allow them to correlate. Parents were 2.04 times more like to provide protection-based explanations for hygiene questions than for the questions about staying inside, $\chi^2$(1, $N$ = 312) = 10.32, $p$ = .001. The opposite was true for social responsibility. Parents were 9.40 times more likely to provide social responsibility explanations for questions about staying inside than for hygiene questions, $\chi^2$(1, $N$ = 312) = 57.15, $p$ < .001. Parents of older children were more likely to provide social responsibility explanation, $OR$ = 1.11, $\chi^2$(1, $N$ = 312) = 6.48, $p$ = .011, and less likely to provide protection-based explanations, $OR$ = 0.90, $\chi^2$(1, $N$ = 312) = 8.37, $p$ = .003, than parents of younger children.

## Shielding

About half of the parents ($n$ = 190, 54.6%) said they do not try to shield their children from information about COVID-19. When parents shielded their children, they did so because they did not want to worry them ($n$ = 126, 36.2%), they thought they were not old enough to understand what is going on ($n$ = 83, 23.8%), they thought their child could not cope with the information ($n$ = 45, 12.9%), they thought people were making too big a deal about the virus ($n$ = 22, 6.3%), or they thought that COVID-19 was not different from a seasonal flu ($n$ = 14, 4.0%). Additionally, a small portion of parents ($n$ = 54, 15.5%) had read or heard advice on how to talk to their children about the pandemic, and the majority found them useful ($n$ = 40, 11.5%).

We added whether parents had seen or heard advice on how to talk to their child about the pandemic to our predictor variables but was not significant. We found that parents of older children were less likely to shield them than parents of younger children, $OR$ = 0.90, $\chi^2$(1, $N$ = 346) = 6.69, $p$ = .010. Additionally, parents who rated their children's biological knowledge higher were less likely to shield them than parents who rated their children's knowledge lower, $OR$ = 0.78, $\chi^2$(1, $N$ = 346) = 7.43, $p$ = .006.

## Discussion

### Content of conversations

Our study examined how parents and children talked about the COVID-19 pandemic. One of our main goals was to examine what type of pandemic-related information children were

seeking. We examined this by looking at the questions parents reported that their children asked. Our results suggest that children asked about changes to their lifestyle brought upon by the pandemic more than they asked about the causes of illness of about how viruses spread. We also examined the responses that parents had to these questions and found that many parents answered these questions directly, and tried to comfort or reassure their child. In their responses parents often gave explanations, appealed to authority, or simply provided facts. This is encouraging as parents that provide more correct explanations have been shown to have children with higher knowledge in that domain [8, 42]. To examine more in-depth the content of parents' responses, we analyzed their responses to a set of predetermined target questions about different aspects of the pandemic. We found that parents' responses to these questions often referenced either protecting the family or social responsibility. Parents also used different types of explanations depending on the question, with parents explaining that the purpose of washing hands was to protect the child, while explaining that their children cannot go to the park because it is their social responsibility to slow the spread of the virus.

We also found that the types of questions children asked were associated with some child characteristics. For example, we found that older children were more likely to ask about issues related to their school or their parents' work. This might be because older children in our sample (7- to 12 year-olds) were more likely to be regularly attending school than the younger children (3- to 5-year-olds). The younger children were also more likely to ask questions about lifestyle changes than older children, which could be because changes like not being able to go outside or see friends might be a great change in their routines. However, these speculations need to be tested in future research. Interestingly, we found that children whose parents reported that the child was coping well with the pandemic and children whose parents reported that the child was very stressed about the pandemic were more likely to ask questions about the virus. The difference between these two groups of children was on the other questions they asked. Children whose parents reported that they were very stressed were also more likely to ask questions about the safety of their family and friends and less likely to ask questions about interpersonal relations and lifestyle changes. Therefore, the children who were seen as more stressed by their parents appear to be focusing only on the virus and its detrimental effects, while the children that were seen as coping well might be asking a lot about the virus but also about more general aspects of the pandemic.

We also found some similarities between the parent-child conversations our participants reported and those reported in prior research. Prior work on parent-child conversations about illnesses, such as the common cold, have shown that parents discuss both why people get sick, but also how to stay healthy [10]. We saw this in our sample, as parents discussed contagion with their children, but also the protective measure they could take to stay healthy. One difference between our findings and prior work on parent-child conversations about illness is that in our sample we found that many of the questions were not about illness. In prior work, the conversations centered on health and illness with families rarely discussing other aspects such as changes to life style [10]. Although lifestyle changes are not unique to a pandemic (e.g., parents could talk about how children have to stay at home when they get a cold), they are probably more salient given the overall disruption to daily routines.

The breadth of children's questions and parental responses we saw in our study is similar to conversations about death, where children ask not only about what causes an organism to die, but also about emotions and what a death means for those who are still living [43]. In our sample, parents reported that children did not only ask about illness, but also about a variety of topics. Likewise, parents not only discussed illness in their answers, but also safety, social norms, and different forms of authority, while also trying to comfort or reassure the child. This suggests that parent-child conversations about the pandemic might be serving multiple

roles, such that they are not only teaching children about illness, but also about social norms and how to cope with stressful situations.

## Factors associated with conversations

A sub goal of our study was to identify variables associated with parent's engagement in conversation and their likelihood of providing explanations. In line with prior research, parents were more likely to provide explanations when their child asked a "how" or "why" question, which request explanations from parents [32, 33]. We also found that the likelihood of providing an explanation did not depend on the age of the child, but parents of younger children were more likely to report questions that requested an explanation. Therefore, parents may provide different information to children of different ages, but this might be due to differences in the questions children are asking. We did find that parents' reports of having sufficient knowledge was associated with whether they ever provided explanations. This result is in line with prior work suggesting that parental attitudes and knowledge are important variables associated with parent-child interactions in a variety of domains including biology [8, 43], social norms [18, 27] and educational opportunities [26, 44, 45].

## Shielding

Given that we were interested in when parents decide to engage in conversations, we also examined parent's reasons for shielding their children from information about the pandemic and thus decide not to engage in conversations with their children about the pandemic. About half of the parents said that they shielded their children from COVID-19 information. These parents tended to say that they shielded their children because they did not want to worry their children or because they thought their children were not old enough to understand the situation. These reasons are also among the top reasons why parents shield their children from information in other domains, such as death [25]. Thus, it appears that the reasons why parents might shield their children from information about COVID-19 are similar to the reasons identified in other domains.

In line with parents' explanations for shielding, we found that shielding was more commonly reported by parents of younger and less knowledgeable children. However, we did not find evidence that parental reports of child coping or stress were related to shielding. This is surprising as parents report not wanting to worry their children as a key reason for shielding them. For older children, parents may generally believe that the children can handle information about the pandemic. Thus, we suggest that shielding as a parental strategy might be more related to parental beliefs about the appropriate age to expose children to information about the pandemic and not on parent's perception of how well their child is coping with the pandemic.

## Limitations

A limitation of our study is that our sample was predominantly white parents. Although not intended, this demographic characteristic might be important as official reports from the Center for Disease Control show that white people have been affected by the pandemic to a lesser extent than other racial groups [46]. Given these facts, the generalizability of our findings should be restricted to white families. Additionally, some of the questions that we asked might have been complicated for parents to answer. Parents might have had different interpretations of what it means to have enough knowledge to answer their children's questions. Therefore, some of our inferential statistics should be interpreted with caution.

Another important limitation of our study is that we do not have actual data of children's questions, only parental reports of these questions. This approach has been used in prior research on children's causal reasoning and death understanding [32, 34], however there are still concerns on whether parents' reporting is accurate. Of particular importance is the retrospective nature of parental reports, as parents were asked to remember the questions their children had asked. It is possible that parents forgot many of the questions children asked or that they were biased towards particular questions that were more common or noteworthy. However, less than a month had passed between the start of the pandemic and when our data was collected, which might lessen, but not completely remove, some of the memory demands of the task. It is also worth noting that because of the pandemic, the results of our study might not speak to parent-child conversations about health and illness more broadly.

## Conclusions

We investigated parental reports of parent-child conversations about the COVID-19 pandemic in an online sample of predominantly white parents from the U.S. We used a mix of qualitative and quantitative methods to identify the topics of their conversations and examined variables associated with parental responses. We found that families primarily discuss information about changes to their lifestyle, preventative measures and the virus. Overall, our study suggests that providing parents with information on how to talk to their children about preventative measures (such as social distancing) and changes in lifestyle, and not just information about how to discuss the virus, might be beneficial as these are the topics children appear to be asking about. Providing parents this information might help as parents that thought they had enough knowledge tended to provide more in-depth answers to their children's questions.

## Supporting information

**S1 File. Supplemental results of stress and coping measures.**
(DOCX)

## Acknowledgments

We would like to thank Dr. Melissa Sturge-Apple, Dr. Sarah Mangelsdorf, Dr. Martha Alibali, Dr. Sarah Brown, and Graciela Trujillo-Hernandez for their comments throughout the research process. We will like to thank the members of the Cognitive Development and Communication lab at the University of Wisconsin-Madison for their support throughout this project.

## Author Contributions

**Conceptualization:** David Menendez, Rebecca E. Klapper, Karl S. Rosengren.

**Data curation:** David Menendez, Rebecca E. Klapper.

**Formal analysis:** David Menendez, Rebecca E. Klapper, Michelle Z. Golden, Ava R. Mandel, Katrina A. Nicholas, Maria H. Schapfel, Olivia O. Silsby, Kailee A. Sowers, Dillanie Sumanthiran, Victoria E. Welch.

**Funding acquisition:** Karl S. Rosengren.

**Investigation:** David Menendez, Karl S. Rosengren.

**Methodology:** David Menendez, Karl S. Rosengren.

**Project administration:** David Menendez, Rebecca E. Klapper.

**Resources:** Karl S. Rosengren.

**Supervision:** David Menendez, Rebecca E. Klapper, Karl S. Rosengren.

**Validation:** David Menendez, Rebecca E. Klapper.

**Visualization:** David Menendez, Rebecca E. Klapper.

**Writing – original draft:** David Menendez.

**Writing – review & editing:** David Menendez, Rebecca E. Klapper, Michelle Z. Golden, Ava R. Mandel, Katrina A. Nicholas, Maria H. Schapfel, Olivia O. Silsby, Kailee A. Sowers, Dillanie Sumanthiran, Victoria E. Welch, Karl S. Rosengren.

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
