## [Decision Letter · Decision Letter 0]

26 Mar 2021

PONE-D-20-40961

"When will it be over?” U.S. Children’s Questions and Parents’ Responses to the COVID-19 Pandemic

PLOS ONE

Dear Dr. Menendez,

Thank you for submitting your manuscript to PLOS ONE. First, let me apologize for delay in receiving this decision letter. During the pandemic, getting reviewers to agree to review manuscripts has been a lengthy process. 

After careful consideration, we feel that your manuscript has merit but does not fully meet PLOS ONE’s publication criteria as it currently stands. Therefore, we invite you to submit a revised version of the manuscript that addresses the points raised during the review process.

Even though one of the reviewers recommended to reject the article, I believe that you can potentially address both reviewer's criticisms to improve the article so it is appropriate for publication. I won't repeat much of their criticisms here, but one of the major themes is a recommendation to consistently describe the data as* parental reports* of children's questions, rather than children's questions. Given you cite work in other domains that has relied on this method (e.g., in understanding death), perhaps you can discuss the degree to which other methods in these domains have corroborated the evidence from parental report with some kind of direct measurement of children. That could help boost confidence in the current findings. In addition if possible, it would be useful to compare these results to domains other than COVID-19, but ideally some other health topic. For example, are the effects of age (and other key predictor variables) on the kinds of questions asked, or explanations offered consistent or inconsistent with other domains of health or safety? Any comparisons of this kind you can make would be helpful.

We look forward to receiving your revised manuscript.

Kind regards,

Micah B. Goldwater, Ph.D

Academic Editor

PLOS ONE

Journal Requirements:

2. We note you have included a table to which you do not refer in the text of your manuscript. Please ensure that you refer to Table 1 in your text; if accepted, production will need this reference to link the reader to the Table.

Reviewers' comments:

Reviewer's Responses to Questions

**Comments to the Author**

1. Is the manuscript technically sound, and do the data support the conclusions?

Reviewer #1: Partly

Reviewer #2: Partly

2. Has the statistical analysis been performed appropriately and rigorously? 

Reviewer #1: I Don't Know

Reviewer #2: I Don't Know

3. Have the authors made all data underlying the findings in their manuscript fully available?

Reviewer #1: Yes

Reviewer #2: Yes

4. Is the manuscript presented in an intelligible fashion and written in standard English?

Reviewer #1: Yes

Reviewer #2: Yes

5. Review Comments to the Author

Reviewer #1: Thank you for this interesting and highly relevant paper. Please find my comments and questions below.

Abstract:

The abstract does provide sufficient informations about the relevance of your study and the main results. I would recommend to include also the goals and research questions.

Introduction:

Perhaps you could structure this part more thoroughly by better separating the information about your work and the current state of the research. The sentence "This information will help us to determine...." could be earlier, as this is important information about the goal of your work.

A more explicit statement of what your work contributes to the current state of research or what open questions it addresses would emphasize the relevance of your findings.

Current study

I miss the elaborated research questions here.

Method

Measures

You should add information about other measures you included in your analysis, such as "biology knowledge" or "stress."

Statistical analysis

I miss assessment criteria for model fit. You should add them here and specify them in the results section.

Results

It is difficult to find the given numbers and percentages in the tables. More guidance is needed here.

Targeted Questions: it is not clear what additional or different information is gained with "targeted questions" compared to "reported questions". Perhaps you could elaborate on this in the first part of this section (and also in the discussion section).

Reporting numbers and percentages: You should standardize this in the paper. Sometimes numbers are reported, sometimes numbers and percentages. High numbers should be written in words (e.g., "two hundred and sixty-five").

More information is needed to assess predictive models: What predictors were included in the model? How does the model fit the data?

Discussion

This section of the article is rather cursory. I would appreciate the summarizing of the findings regarding the research questions. What are the main results and what do they add regarding to the research question, the state of research and the advice parents should get on the topic.

Reviewer #2: The submitted manuscript reports on an online study of parents asked to report on the pandemic-related questions asked by their children and the ways in which the parents responded to them. The manuscript is primarily descriptive, reporting the ways in which children’s questions varied by age as well as the different themes parents used when answering them. The conclusions made in the Discussion go beyond the collected data, in my opinion, as associations were not made among any of the collected data about questions or responses and other measures of well-being during the pandemic (although relevant data were collected; see Table 1). My other comments on this paper are found below.

MAJOR CONCERNS

1. Prolific Academic, another online recruitment platform used in psychology studies, requires potential participants to complete screening questions that are used to determine which studies are assigned to them (this determination is made along with the recruitment criteria specified by the researchers). These same procedures are not followed in Amazon Mechanical Turk and, as such, there is some concern that participants may carefully read the presented consent form, identify the characteristics of the sample under investigation, and correctly answer in-study screening questions not because they actually fit the demographic, but because they indicated that they are a member of the target demographic (even though that may not be the case). To what extent is this situation a concern with this sample? Did the consent form list the specific criteria used to recruit participants so that they could have accurately answered the screening questions in the absence of having actual children? Were there any duplicate responses in the data (e.g., parents who completed the questionnaire more than once)? What confidence do the researchers have that the participants were actually parents of children in the specified age ranges?

2. Some of the data reported in the manuscript seem somewhat unknowable with any measureable level of confidence. Anecdotally, as a parent of two children in the specified age ranges, I could not say with any confidence what percent of conversations I have had with my children over the past month were initiated by me or by my child with the most recent birthday. Do the researchers have any measures of confidence provided by the parents regarding who started the conversation, how parents responded to children’s requests, and so on? Similarly, asking parents whether they have enough biology knowledge to answer their children’s questions is complicated – parents may not know enough about biology to know what they do not know.

3. The authors indicate that the qualitative coding scheme was developed based on the data from the 30% of participants who did not pass the attention checks. Were these data different in kind in any way from those provided by parents who did pass the attention checks (e.g., demographic characteristics, number of questions asked by children, and so on)? It may have been better to develop the coding scheme based on pilot data or on a subset of the useable data, with plans to go back and recode those data at the end of the study when the scheme was well-established.

4. The language used throughout the manuscript has to be more carefully tailored to the actual method used (e.g., the statement “older children were more likely to ask questions about… safety” should be revised to “parents reported that older children more frequently asked questions about safety”). Similarly, in multiple instances, the authors make comparative statements without clearly referencing the comparison group (e.g., in the previous statement, it is unclear whether older children were more likely to ask questions about safety relative to younger children or relative to some other type of question).

5. The Discussion section goes beyond the data by making recommendations as to how parents should talk with children about the pandemic in the absence of data indicating whether the type of questions asked and answered is associated with well-being (e.g., “providing parents with information on how to talk to their children about preventative measures (such as social distancing) and changes in lifestyle, and no just information about how to discuss the virus, would be beneficial and children typically ask about them”). Interestingly, however, measures of anxiety, stress, and coping are presented in Table 1 but are not described in the manuscript. These could be important correlates of the types of questions children ask and parents answer, but this information is not provided in the manuscript.

MINOR CONCERNS

6. The abstract should not reference advice columns, as this is not a scientific source.

7. In the introduction, the authors indicate that they examine parent-child conversations from a developmental perspective. The authors do not directly examine parent-child conversations, but instead report on questions parents say their children asked them and how the parents responded. Second, the authors use a cross-sectional perspective based on child age, not a developmental perspective (which implies study of the same children over time).

8. The attention check questions should be included in the manuscript.

9. The options parents were provided about the reasons they shielded their children from information should be provided in the text.

10. The statement indicating that “some parents also reported just an overall increase in the number of questions while others reported a decrease in questions” would be much more informative if the percentages associated with each position were included as well.

11. The statement that “parents were 14.2 times more likely to provide an explanation when the child’s question elicited one” is circular.

12. Table 2 indicates that questions coded in the “elicits explanation” category could also be coded in other categories, an important detail that should be mentioned in the text.

13. The manuscript would benefit from some reformatting, including the use of page numbers and line numbers as well as consistency in the presentation of references in the text of the paper. Spelling and punctuation errors should also be corrected, particularly in the qualitative examples provided in the text.

6. PLOS authors have the option to publish the peer review history of their article (what does this mean?). If published, this will include your full peer review and any attached files.

Reviewer #1: **Yes: **Anita Sandmeier

Reviewer #2: No

---

## [Author Response · Author response to Decision Letter 0]

12 May 2021

Editor:

one of the major themes is a recommendation to consistently describe the data as parental reports of children's questions, rather than children's questions. 

We have made this change throughout the paper.

Given you cite work in other domains that has relied on this method (e.g., in understanding death), perhaps you can discuss the degree to which other methods in these domains have corroborated the evidence from parental report with some kind of direct measurement of children. That could help boost confidence in the current findings. 

We added a new section to the introduction that examines the use of parental reports in developmental science. Please see the new section in page 4-5.

In addition if possible, it would be useful to compare these results to domains other than COVID-19, but ideally some other health topic. For example, are the effects of age (and other key predictor variables) on the kinds of questions asked, or explanations offered consistent or inconsistent with other domains of health or safety? Any comparisons of this kind you can make would be helpful.

We have expanded our discussion section and now include some comparisons between our findings and those in prior work in parent-child conversations. We specifically discuss how our findings on the types of questions children ask and the answers parents provide about the pandemic relate to prior work on parent-child conversations about illness, but also conversations in other domains such as death. Please see the revised discussion section.

Reviewer #1: 

Thank you for this interesting and highly relevant paper. Please find my comments and questions below.

Abstract:

The abstract does provide sufficient information about the relevance of your study and the main results. I would recommend to include also the goals and research questions.

We have now added our research questions to the abstract.

Introduction:

Perhaps you could structure this part more thoroughly by better separating the information about your work and the current state of the research. The sentence "This information will help us to determine...." could be earlier, as this is important information about the goal of your work.

We thank the reviewer for their comment. We have restructured parts of the introduction to address the reviewer’s issues.

A more explicit statement of what your work contributes to the current state of research or what open questions it addresses would emphasize the relevance of your findings.

We have added the following statement to page 5-6 to help make the contribution of our study clearer, “Therefore, this study provides information about the type of information children are seeking about illness, which can enhance our understanding of the development of illness concepts. In addition, this study provides insight in how parents answer these questions and the variety of the information they provide to their children.”

Current study

I miss the elaborated research questions here.

We thank the reviewer for pointing out this oversight. We have made our research questions more explicit in this section. In page 5 we added “The main question this study is investigating is what kind of information related to the pandemic are children seeking and what type of information are parents providing them?” 

Method

Measures

You should add information about other measures you included in your analysis, such as "biology knowledge" or "stress."

We have added information about these measures to the method section.

Statistical analysis

I miss assessment criteria for model fit. You should add them here and specify them in the results section.

We have added this information to the statistical analysis section. We now report the results of the Hosmer and Lemeshow goodness of fit test to provide some basis as to whether our model fits the data. We note that because this approach relies on p-values, the likelihood of the model fitting the data increases as sample size increases. For this reason, we also include information about how regardless of model fit, we use Wald’s test to determine if our predictors add explanatory power to our models.

Results

It is difficult to find the given numbers and percentages in the tables. More guidance is needed here.

We attempted to make the reporting of the different values more clearly throughout the paper. We now make reference to the appropriate table closer to the beginning of the paragraph. We also report some of the numbers from the table in the body of the text in order to make it easier for people to find the appropriate number.

Targeted Questions: it is not clear what additional or different information is gained with "targeted questions" compared to "reported questions". Perhaps you could elaborate on this in the first part of this section (and also in the discussion section).

We thank the reviewer for this comment. We tried to clarify the importance of the targeted questions in both the method and results section. See pages 7-8 and 13.

Reporting numbers and percentages: You should standardize this in the paper. Sometimes numbers are reported, sometimes numbers and percentages. High numbers should be written in words (e.g., "two hundred and sixty-five").

We have tried be as consistent as possible in how we report numbers throughout the revised paper.

More information is needed to assess predictive models: What predictors were included in the model? How does the model fit the data?

We have clarified each of the models in the paper. We have tried to make clearer the type of model we ran and the predictors we included. Results for the model fit are presented in Table 6.

Discussion

This section of the article is rather cursory. I would appreciate the summarizing of the findings regarding the research questions. What are the main results and what do they add regarding to the research question, the state of research and the advice parents should get on the topic.

We have expanded our discussion section in order to tie our results to more to prior literature and provide more information about our research questions. We have also revised the discussion in light of reviewer 2’s concerns. Please see the revised discussion section.

Reviewer #2: 

The submitted manuscript reports on an online study of parents asked to report on the pandemic-related questions asked by their children and the ways in which the parents responded to them. The manuscript is primarily descriptive, reporting the ways in which children’s questions varied by age as well as the different themes parents used when answering them. The conclusions made in the Discussion go beyond the collected data, in my opinion, as associations were not made among any of the collected data about questions or responses and other measures of well-being during the pandemic (although relevant data were collected; see Table 1). My other comments on this paper are found below.

MAJOR CONCERNS

1. Prolific Academic, another online recruitment platform used in psychology studies, requires potential participants to complete screening questions that are used to determine which studies are assigned to them (this determination is made along with the recruitment criteria specified by the researchers). These same procedures are not followed in Amazon Mechanical Turk and, as such, there is some concern that participants may carefully read the presented consent form, identify the characteristics of the sample under investigation, and correctly answer in-study screening questions not because they actually fit the demographic, but because they indicated that they are a member of the target demographic (even though that may not be the case). To what extent is this situation a concern with this sample? Did the consent form list the specific criteria used to recruit participants so that they could have accurately answered the screening questions in the absence of having actual children? Were there any duplicate responses in the data (e.g., parents who completed the questionnaire more than once)? What confidence do the researchers have that the participants were actually parents of children in the specified age ranges?

We thank the reviewer for this comment. We took many different measures to make sure that our participants were parents. First, Mechanical Turk allows researchers to make their studies available only to people with certain qualifications, and parenthood status is one of the possible options. We made sure that only participants who were marked as parents in their system could access the survey. In addition, participants completed a screening survey BEFORE seeing the consent form. In this screening survey there were questions about being a parent, and having a child in the appropriate age range for the study. These questions were embedded within other distractor questions to make it more difficult for potential participants to guess which of the questions were the target ones. Furthermore, we asked participants to report the age of all of their children at the beginning and then report the age of the child they were focusing on again at the end of the survey. This was with the idea that participant who might have been lying about being parents would have a difficult time remembering the age they said at the beginning of the survey. Participants whose reported ages did not match were excluded from analysis (we allowed a difference of 1 year in case participants accidentally mistype). Finally, Qualtrics allows an option to “prevent ballot stuffing” which prevents people with the same IP address from completing the study. This option decreases the likelihood that participants provided multiple responses. We have included this information in the method section. Please see pages 6-9.

2. Some of the data reported in the manuscript seem somewhat unknowable with any measureable level of confidence. Anecdotally, as a parent of two children in the specified age ranges, I could not say with any confidence what percent of conversations I have had with my children over the past month were initiated by me or by my child with the most recent birthday. Do the researchers have any measures of confidence provided by the parents regarding who started the conversation, how parents responded to children’s requests, and so on? Similarly, asking parents whether they have enough biology knowledge to answer their children’s questions is complicated – parents may not know enough about biology to know what they do not know.

We agree with the reviewer and have added this issue as a limitation of our study. This is an issue that could really influence some of the inferential statistics performed on out data. Please see the revised limitation section in pages 24-25. While we agree with the reviewer that it may be difficult to remember with the child or oneself initiated certain questions we would argue that questions about the pandemic, similar to questions about death, are quite salient and may be more memorable than “every day conversations” with children enabling many parents to provide a rough estimate regarding the percentage of conversations started by the child. 

3. The authors indicate that the qualitative coding scheme was developed based on the data from the 30% of participants who did not pass the attention checks. Were these data different in kind in any way from those provided by parents who did pass the attention checks (e.g., demographic characteristics, number of questions asked by children, and so on)? It may have been better to develop the coding scheme based on pilot data or on a subset of the useable data, with plans to go back and recode those data at the end of the study when the scheme was well-established.

We understand the reviewer’s concerns. The data from the 30% of participants who did not pass the attention checks did have some differences from the parents who did pass them, however, these differences were mostly on the number of idiosyncratic responses (which we coded in the “other” category). We decided it was better to use the discarded data to develop our coding scheme before moving to the actual data, this way our reliability estimate would not be inflated by coders have prior experience with the data. As we coded the data of participants who passed the attention checks we looked for responses that might have not fit our coding scheme or for new categories to add, but we did not find any such instances. This suggests that the coding scheme developed with the data of parents who failed the attention check captures well captures the responses of the parents who did pass the attention check pretty well. We now include this information in pages 9-10.

4. The language used throughout the manuscript has to be more carefully tailored to the actual method used (e.g., the statement “older children were more likely to ask questions about… safety” should be revised to “parents reported that older children more frequently asked questions about safety”). Similarly, in multiple instances, the authors make comparative statements without clearly referencing the comparison group (e.g., in the previous statement, it is unclear whether older children were more likely to ask questions about safety relative to younger children or relative to some other type of question).

We have cleaned up our language throughout the revised manuscript.

5. The Discussion section goes beyond the data by making recommendations as to how parents should talk with children about the pandemic in the absence of data indicating whether the type of questions asked and answered is associated with well-being (e.g., “providing parents with information on how to talk to their children about preventative measures (such as social distancing) and changes in lifestyle, and no just information about how to discuss the virus, would be beneficial and children typically ask about them”). Interestingly, however, measures of anxiety, stress, and coping are presented in Table 1 but are not described in the manuscript. These could be important correlates of the types of questions children ask and parents answer, but this information is not provided in the manuscript.

We have added our measures of stress and coping to our analyses. Given that we added these two variables, we have updated the results throughout the results section. Most of the findings stayed the same, but a couple are no longer significant. Please see the revised results. We have also revised the discussion section to stay closer to our actual data.

MINOR CONCERNS

6. The abstract should not reference advice columns, as this is not a scientific source.

We have deleted this information from the abstract.

7. In the introduction, the authors indicate that they examine parent-child conversations from a developmental perspective. The authors do not directly examine parent-child conversations, but instead report on questions parents say their children asked them and how the parents responded. Second, the authors use a cross-sectional perspective based on child age, not a developmental perspective (which implies study of the same children over time).

We have made the appropriate changes to this section in order to make our language match our methods. In page 3, we now state “In this paper, we examine parent-child conversations about COVID-19 from a developmental psychology perspective. Specifically, we examine parental reports of children’s questions and their answers.” We kindly disagree with the reviewer’s second point. We agree that our study is cross-sectional rather than longitudinal, but we disagree that the only way to have a developmental perspective is by examining children longitudinally. Many studies in developmental psychology, and the majority of those in the field of cognitive development, have taken a cross-sectional approach in order to understand development. We include a paragraph in pages 4-5 that discusses the use of our methodology in previous developmental psychology work.

8. The attention check questions should be included in the manuscript.

We have included this information in the method section.

9. The options parents were provided about the reasons they shielded their children from information should be provided in the text.

We have added this information to the method section.

10. The statement indicating that “some parents also reported just an overall increase in the number of questions while others reported a decrease in questions” would be much more informative if the percentages associated with each position were included as well.

We previously reported these numbers in Table 3, but we now also report them in the body of the text.

11. The statement that “parents were 14.2 times more likely to provide an explanation when the child’s question elicited one” is circular.

We do not feel that this statement is circular but believe that the confusion comes from the naming of our coding categories. The category for “eliciting explanation” was used when children asked questions they asked for explanations, such as why or how questions. We have changed the name of this category to “request explanations” to hopefully avoid this confusion. The sentence the reviewer highlighted now reads “parents were 14.2 times more likely to provide an explanation when the child’s question requested one.” Rather than being circular, we believe this finding shows that that although some parents who did not provide an explanation even when children asked for one, parents were generally providing the information that children requested. We have made this clearer in the manuscript. 

12. Table 2 indicates that questions coded in the “elicits explanation” category could also be coded in other categories, an important detail that should be mentioned in the text.

We agree with the reviewer. We now mention this information in the coding section of the method.

13. The manuscript would benefit from some reformatting, including the use of page numbers and line numbers as well as consistency in the presentation of references in the text of the paper. Spelling and punctuation errors should also be corrected, particularly in the qualitative examples provided in the text.

We apologize for this oversight, we have double checked the manuscript for stylistic, grammatical and orthographical errors. However, we do not think it is appropriate for us to edit quotes provided by our participants, as these quotes are the actual data we used in our study. We have added that this information in page 11.

---

## [Decision Letter · Decision Letter 1]

5 Jul 2021

PONE-D-20-40961R1

"When will it be over?” U.S. Children’s Questions and Parents’ Responses about the COVID-19 Pandemic

PLOS ONE

Dear Dr. Menendez,

Thank you for submitting your manuscript to PLOS ONE. After careful consideration, we feel that it has merit but does not fully meet PLOS ONE’s publication criteria as it currently stands. Therefore, we invite you to submit a revised version of the manuscript that addresses the points raised during the review process. 

I apologize for the delay to receive this decision. I had hoped that both of the original reviewers would review the manuscript again, but decided it was not right to keep you waiting further when you could otherwise be working to address the reviewer's comments who did submit a second review. I also note that you adequately addressed the points I brought up in the first round in this revision, so please just focus on the sole reviewer's comments. 

We look forward to receiving your revised manuscript.

Kind regards,

Micah B. Goldwater, Ph.D

Academic Editor

PLOS ONE

Journal Requirements:

Reviewers' comments:

Reviewer's Responses to Questions

**Comments to the Author**

1. If the authors have adequately addressed your comments raised in a previous round of review and you feel that this manuscript is now acceptable for publication, you may indicate that here to bypass the “Comments to the Author” section, enter your conflict of interest statement in the “Confidential to Editor” section, and submit your "Accept" recommendation.

Reviewer #1: (No Response)

2. Is the manuscript technically sound, and do the data support the conclusions?

Reviewer #1: Partly

3. Has the statistical analysis been performed appropriately and rigorously? 

Reviewer #1: No

4. Have the authors made all data underlying the findings in their manuscript fully available?

Reviewer #1: Yes

5. Is the manuscript presented in an intelligible fashion and written in standard English?

Reviewer #1: Yes

6. Review Comments to the Author

Reviewer #1: Thank you for adressing the points of my first feedback.

I have still concerns regarding the following points:

Statistical analysis and criteria for model fit:

I suggested to include model fit in ordert o test, if the measurement models of coping and stress fit the data well. This is reported with goodness of fit values such as the χ2/df (ratio chi-square and degrees of freedom), CFI (comparative fit index), the RMSEA (root mean square error of approximation), and the SRMR (Standardised Root Mean Square Residual). CFI values of .90 or greater indicate adequate, CFI values of .95 and greater indicate good fit. RMSEA values of .08 or less and .05 or less indicate adequate and good fit, respectively (Hu & Bentler, 1999). I found these measurement models in your R code, but not the documention of the values in the paper.

You instead reported the Hosmer and Lemeshow goodness of fit test (table 6), that tests the goodness of fit of logistic regression models. You applied Generalized linear mixed-effects models (with a binomial link function). As to my knowledge GLMM need other goodness-of-fit methods (https://www.frontiersin.org/articles/10.3389/fpsyg.2021.666182/full). . In my opinion this is crucial to assess the quality of your models where you try to «predict» some outcomes.

Regarding the wording I have an additional concern: You have only cross-sectional data that doesn’t allow to «predict» an outcome.

On this background I assess the statistical analysis as not appropriate and rigorous enough.

Discussion

This section is still confusing as you do not answer your research questions in a structured way. The distinction between reported and targeted questions is not mentioned and you report results, that are not mentioned before (p. 24, line 437 – 439). This part needs a rigorous revision that stringently and logically puts the results in relation to each other.

7. PLOS authors have the option to publish the peer review history of their article (what does this mean?). If published, this will include your full peer review and any attached files.

Reviewer #1: No

---

## [Author Response · Author response to Decision Letter 1]

21 Jul 2021

Reviewer #1: 

Thank you for adressing the points of my first feedback.

We thank the reviewer for their positive feedback and their thorough read of our manuscript.

I have still concerns regarding the following points:

Statistical analysis and criteria for model fit:

I suggested to include model fit in order to test, if the measurement models of coping and stress fit the data well. This is reported with goodness of fit values such as the χ2/df (ratio chi-square and degrees of freedom), CFI (comparative fit index), the RMSEA (root mean square error of approximation), and the SRMR (Standardised Root Mean Square Residual). CFI values of .90 or greater indicate adequate, CFI values of .95 and greater indicate good fit. RMSEA values of .08 or less and .05 or less indicate adequate and good fit, respectively (Hu & Bentler, 1999). I found these measurement models in your R code, but not the documention of the values in the paper.

Given that this measurement model is not the main topic of our paper, we provide this information in the supplemental materials (S1 file). In this file, we previously only included fit indices for the structural equation model predicting the factor scores. We did not report fit indices on the factor analysis as this analysis was exploratory, and so we did not want to give the impression that we were testing how well a theoretically motivated model fit the data. To attempt to address the reviewer’s concern we have clearly labelled the explanatory factor analysis as such in the paper and the supplemental materials. Then, we detail how we fitted a confirmatory factor analysis in order to see how well the exploratory factor analysis fitted the data, and report these values. Because the measurement of stress and coping is not the main concern of our paper, we include all of this information in the supplemental materials. We apologize for misunderstanding which model the reviewer was referring to in their first review.

You instead reported the Hosmer and Lemeshow goodness of fit test (table 6), that tests the goodness of fit of logistic regression models. You applied Generalized linear mixed-effects models (with a binomial link function). As to my knowledge GLMM need other goodness-of-fit methods (https://www.frontiersin.org/articles/10.3389/fpsyg.2021.666182/full). In my opinion this is crucial to assess the quality of your models where you try to «predict» some outcomes.

We report the Hosmer and Lemeshow test as it provides a chi-square for how the model fits the data by comparing the number of observations the model expects to the observed observations. This is a test often used for logistic regression models (Hosmer, Lemeshow, & Sturdivant, 2013). The paper the reviewer references suggest that many articles report the AIC. We previously did not report the AIC as this statistic is not interpretable on its own. AIC is only meaningful in model selection, where it can be used as a criterion for selecting between different models. Given that we are not comparing different models in our paper, the AIC is not interpretable. In order attempt to address the reviewers’ concerns we now also report the AIC and BIC in Table 6, but do not attempt to interpret them.

Hosmer DW, Lemeshow S, Sturdivant RX. Applied Logistic Regression, 3rd Edition. 2013. New York, USA: John Wiley and Sons.

Regarding the wording I have an additional concern: You have only cross-sectional data that doesn’t allow to «predict» an outcome.

We have revised the wording throughout the manuscript and now use “associated with” instead of “predict” when relevant.

On this background I assess the statistical analysis as not appropriate and rigorous enough.

We hope some of the changes we made highlight the appropriateness and rigor of our analyses.

Discussion

This section is still confusing as you do not answer your research questions in a structured way. 

We have attempted to provide more structure to the discussion section. We have reorganized the information and added subheadings to attempt to provide more structure. We also added more discussion of some of the results in order to relate the findings to each other (last comment by the reviewer). Please see revised discussion section.

The distinction between reported and targeted questions is not mentioned 

We now mention that we had both reported and targeted questions, and the reason why we examined each. We also discuss the results of both types of questions and clearly label which we are discussing. 

and you report results, that are not mentioned before (p. 24, line 437 – 439). 

This information was included in the results section. In page 24 of the previous manuscript, we said “Additionally, shielding does not seem related to how the child is coping with the pandemic, but rather to parents’ beliefs about the appropriate age to expose children to this information.” In the results section (page 23), we mention how many parents reported shielding their children because “they thought they were not old enough to understand what is going on (n = 83, 23.8%)” (line 402-403). In that same page we also mention how our other predictors, including child coping did not predict shielding. We have attempted to clarify this information on the discussion section.

This part needs a rigorous revision that stringently and logically puts the results in relation to each other.

We have attempted to revise the discussion section as suggested by the reviewer. We hope the new discussion section addresses the reviewer’s concerns and shows the novelty of our findings.

---

## [Decision Letter · Decision Letter 2]

13 Aug 2021

"When will it be over?” U.S. Children’s Questions and Parents’ Responses about the COVID-19 Pandemic

PONE-D-20-40961R2

Dear Dr. Menendez,

We’re pleased to inform you that your manuscript has been judged scientifically suitable for publication and will be formally accepted for publication once it meets all outstanding technical requirements.

Kind regards,

Micah B. Goldwater, Ph.D

Academic Editor

PLOS ONE

Additional Editor Comments (optional):

Reviewers' comments:

Reviewer's Responses to Questions

**Comments to the Author**

1. If the authors have adequately addressed your comments raised in a previous round of review and you feel that this manuscript is now acceptable for publication, you may indicate that here to bypass the “Comments to the Author” section, enter your conflict of interest statement in the “Confidential to Editor” section, and submit your "Accept" recommendation.

Reviewer #1: All comments have been addressed

2. Is the manuscript technically sound, and do the data support the conclusions?

Reviewer #1: Yes

3. Has the statistical analysis been performed appropriately and rigorously? 

Reviewer #1: Yes

4. Have the authors made all data underlying the findings in their manuscript fully available?

Reviewer #1: Yes

5. Is the manuscript presented in an intelligible fashion and written in standard English?

Reviewer #1: Yes

6. Review Comments to the Author

Reviewer #1: Dear authors

thank you for adressing my concerns of the second round of review and thank you for your explanations regarding model fit measures that are very plausible. I'm looking forward to see your interesting paper published.

Best wishes, Reviewer #1

7. PLOS authors have the option to publish the peer review history of their article (what does this mean?). If published, this will include your full peer review and any attached files.

Reviewer #1: No

---

## [Editor Report · Acceptance letter]

17 Aug 2021

PONE-D-20-40961R2 

“When will it be over?” U.S. children’s questions and parents’ responses about the COVID-19 pandemic 

Dear Dr. Menendez:

I'm pleased to inform you that your manuscript has been deemed suitable for publication in PLOS ONE. Congratulations! Your manuscript is now with our production department. 

Kind regards, 

on behalf of

Dr. Micah B. Goldwater 

Academic Editor

PLOS ONE